# Enhancing Medical Students’ Knowledge and Performance in Otolaryngology Rotation through Combining Microlearning and Task-Based Learning Strategies

**DOI:** 10.3390/ijerph20054489

**Published:** 2023-03-03

**Authors:** Farzaneh Sedaghatkar, Aeen Mohammadi, Rita Mojtahedzadeh, Roghayeh Gandomkar, Mahtab Rabbani Anari, Sasan Dabiri, Ardavan Tajdini, Sepideh Zoafa

**Affiliations:** 1Department of Medical Education, School of Medicine, Tehran University of Medical Sciences, Tehran 1416633591, Iran; 2Department of E-Learning in Medical Education, Center of Excellence for E-Learning in Medical Education, School of Medicine, Tehran University of Medical Sciences, Tehran 1416614741, Iran; 3Health Professions Education Research Center, Tehran University of Medical Sciences, Tehran 1416633591, Iran; 4Otorhinolaryngology Research Center, AmirAlam Hospital, Tehran University of Medical Sciences, Tehran 1145765111, Iran

**Keywords:** microlearning, task-based learning, medical education, knowledge, clinical performance

## Abstract

Microlearning is recommended to be implemented within the context of a wider teaching–learning ecosystem, especially in real working environments. Task-based learning is used in clinical education setting. This study aims at assessing the effect of an integrated approach of microlearning with task-based learning on medical students’ knowledge and performance in Ear, Nose and Throat clerkship rotation. A total of 59 final-year medical students participated in this quasi-experimental study which included two control groups (routine teaching and task-based learning methods) and one intervention group (combined microlearning and task-based learning method). Pre- and post-tests of students’ knowledge and performance were assessed through a multiple-choice question test and a Direct Observation Procedural Skills (DOPS) instrument, respectively. Performing Analysis of Covariance for knowledge post-test scores among three groups revealed significant differences (F = 3.423, *p*-value = 0.040) and the intervention group had the highest score. Analyzing DOPS results showed that the intervention group achieved significantly higher scores compared to the control ones for all the expected tasks (*p*-values = 0.001). The findings of the present study show that the combined strategy of microlearning with task-based learning is an effective clinical teaching method for enhancing medical students’ knowledge and performance in a real working environment.

## 1. Introduction

Generation Z medical students, born between 1995 and 2012, have specific learning preferences and seek personalized learning opportunities that help them achieve optimal use of time and resources. They enjoy short activities and desire to have access to their learning needs independently in the moment with the help of technology, especially while performing tasks. In addition, they prefer to receive feedback on their performance just in time [1]. Hence, clinical teachers have to implement on-the-job learning experiences that are technology-enhanced and need short attention spans [1,2]. Microlearning (ML) is one of the strategies that is recommended for generation Z medical students [2], because it addresses their preferences with the help of technology [3]. As these students are not engaged with long lectures or time-consuming learning activities [2], the best way is to provide them with meaningful and concise chunks of the material that are quickly and easily learned [4]. The duration of these instructional units ranges from a matter of seconds to 15 min [5], and can be in the form of micro videos, job aids, quizzes, assignments and case studies [6]. The benefits of implementing ML include improved concepts’ retention, higher students’ motivation, more engagement in collaborative learning and better students’ performance. Considering these advantages, ML is widely studied in health professional education [7] and it has been shown that this strategy enhances learners’ engagement and has positive effects on their knowledge and confidence while performing clinical tasks. Furthermore, it is a foundation for promoting critical thinking and clinical reasoning. However, despite these benefits, ML is not a suitable strategy to be implemented alone for teaching complicated subjects to health professional students [5], and it should be applied within the context of a wider teaching–learning ecosystem [8], especially while teaching a specific task in working environments [7] or a wide range of information [9]. In these cases, a blended approach of ML and other teaching strategies is preferred [7,9,10].

Task-Based Learning (TBL) is a clinical teaching–learning strategy that is suitable to be combined with ML. In TBL, students visit the patients in a real clinical setting and are guided to learn the related tasks through understanding the underlying concepts and mechanisms and then applying the acquired knowledge and skills in other situations [11]. Indeed, TBL focuses on not only performing the task, but also understanding the relevant basic and clinical medical knowledge, and moreover developing generic competencies such as communication skills and problem solving [12]. In this way, TBL supports the integration of medical knowledge with patient care, i.e., amalgamation of the theory and practice [13]. It is shown that TBL is an effective clinical teaching strategy for both undergraduate [14,15] and post graduate [16] medical education. However, there are some potential limitations for implementing TBL in medical education. In TBL, the teaching topics and tasks are not systematically structured and organized. Therefore, it is difficult for some students, especially ones with poor self-directed learning abilities, to gain a comprehensive understanding of the task performance. Hence, it is recommended to modify TBL strategy with a focus on adopting suitable methods for imparting the prerequisite knowledge to students while they are conducting the tasks [17], for which ML can be an appropriate strategy.

Thus, regarding the emerging trend of using ML in clinical education and the recommendation of integrating it with work-based learning strategies on one hand and the need to modify TBL for better knowledge provision to the students on the other hand, we assessed the effect of a combined ML–TBL method on final-year medical students’ knowledge and performance for the selected tasks in an otolaryngology rotation.

## 2. Materials and Methods

This quasi-experimental study with non-equivalent pre- and post-test design included two control and one intervention groups, and was conducted from October 24 to December 19, 2021. The study was approved by the Ethical Committee of Tehran University of Medical Sciences (Reference number: ID.IR.TUMS.MEDICINE.REC.1399.608).

### 2.1. Participants and Setting

A total of 59 final-year medical students in their clerkship rotation to the otolaryngology ward of an educational hospital participated in the study. The hospital is an otolaryngology referral center which hosts a wide variety of patients with a range of conditions, from simple conditions to the most complicated ones. Medical students spend two weeks in this rotation, particularly in clinics and the emergency unit, where they are the first line encounters to the patients. They have to manage the patients and perform routine clinical procedures under the supervision of residents and clinical teachers. In fact, before conducting this study, the teaching method in this rotation was based on learning to conduct assigned tasks; however, it was not designed as a TBL method to ensure students’ learning. Routinely about 10 students are allocated to each rotation. Therefore, we assigned two rotations to each of the study groups, i.e., 20, 20 and 19 students to the first control, second control and intervention groups, respectively. We briefed students about the study purpose and design, and obtained their informed consent. They had the right to withdraw their participation in the study at any time. Meanwhile, they were assured that this decision would not affect their learning and assessment experience in the rotation.

### 2.2. Preparation Phase

Two medical educationists held a three-hour workshop for three otolaryngology clinical teachers to brief them on the study design, TBL, ML and teaching–learning process in each of the study groups. After this orientation, the otolaryngology clinical teachers reviewed the curriculum and selected five essential clinical tasks to be covered in the study. They considered the following criteria for selecting the tasks: (a) the task was either a common problem in the community or a complicated one; (b) the task was likely to be encountered by students during their clinical rotation; and (c) the task could be trained through TBL and ML. The final selected tasks were ear examination, ear irrigation, subcuticular suture, nasal packing and epistaxis management. They devised the learning objectives for each of these tasks.

Then, the clinical teachers created a micro-video for each of the above-mentioned tasks. For this purpose, they first worked on the scenario of each video in order to concisely cover the main objectives, task steps and students’ common mistakes. The scenarios were reviewed by another clinical teacher to ensure the appropriateness of the content and the coverage of the learning objectives. Moreover, an e-learning specialist helped the clinical teachers in consideration of pedagogical aspects while creating the contents. The videos included the performance of the task on real patients or manikins alongside with the narration of the clinical teachers focusing on the main points. Whenever necessary, a few slides or images were also included. Each of the five videos was 7 to 10 min long with an average of 8 min.

### 2.3. Instruments

We used two instruments to assess the students’ knowledge and performance:For assessing the knowledge, we developed a 15-item multiple-choice question test with maximum score of 15. The test covered the intended learning objectives related to the five selected tasks in different levels of Blooms taxonomy, i.e., taxonomy one (knowledge and understanding), taxonomy two (application and analysis) and taxonomy three (synthesis and evaluation) [18]. The test included one case-based scenario for each of the tasks. Two other otolaryngology clinical teachers who were not involved in the study confirmed the test with regard to the coverage of the objectives. The test reliability was 0.76 using the Kuder–Richardson 20 formula which has been proven to be acceptable [19].To assess students’ performance, we used the Direct Observation of Procedural Skills (DOPS) assessment method. DOPS is a workplace-based assessment method for evaluating students’ procedural skills [20]. It encourages a deep approach to learning through helping students identify their areas of weaknesses and improve their performance. Evaluating each clinical procedure or task needs a standard DOPS checklist [21]. In this study, we developed a separate checklist for each of the selected tasks which included 5, 6, 7, 10 and 5 items for ear examination, ear irrigation, epistaxis management, subcuticular suture and nasal packing tasks respectively. These checklists covered relevant items to each task such as awareness of the anatomy, pre-procedure preparation, compliance with sterile principles, performance of the task steps, post-procedure actions, communication skills, obtaining patient consent and complying with medical ethics. In addition, there was a question asking for rating the student’s overall performance. Each individual student performance on each item was rated by an observer, a clinical teacher or resident, within a five Likert type scale continuum consisting of “couldn’t perform (incompetent)”, “less than expected (needs further development)”, “moderate”, “acceptable (competent enough)” and “higher than expected (excellent)”. The tool’s content validity was checked by three otolaryngologists. The reliability was confirmed by the Cronbach’s alpha of 0.841 [19].

### 2.4. Study Design

To avoid contamination of the study groups, we conducted the study in the first control group, followed by the second control and intervention groups. Students in the otolaryngology rotation routinely participate in an orientation session at the beginning of their rotation to be briefed on the expected competencies, duties and tasks. In this session, we explained the study aims, obtained students’ informed consent and took the knowledge pre-test. Participants of each study group underwent their assigned teaching–learning method; namely, routine, TBL or combined ML–TBL. On the last day of the rotation, they once again took the knowledge test as the post-test. In addition, during the rotation, clinical teachers and residents filled the DOPS tool for each individual student performing each task and provided them with feedbacks on their performance. In the following, we explain the teaching–learning strategy in each of the study groups.

Routine teaching–learning method in the first control group: Students in this group experienced the routine program of the otolaryngology ward. They were responsible for visiting patients and performing routine procedures and tasks, including five selected tasks for this study, in the clinics and emergency unit. Clinical teachers and residents supervised students and were accessible for answering their questions and guiding them for patient management. However, with regard to the continuum of systematic to opportunistic clinical program in the SPICES model [22], there was no systematic approach to monitor the variety of patients visited by students. They managed only those patients that opportunistically referred to the clinics and emergency unit. In addition, students attended some lecture-based classes on a variety of clinical conditions according to the curriculum.TBL method in the second control group: In this study group, after briefing the students about TBL, they received a study guide through WhatsApp social media including the list of tasks, their related learning objectives and the important points for performing the tasks. The students visited and managed the patients like the previous study group. Meanwhile, students attended a 20- to 30-min interactive lecture-based class for each task at the middle of the rotation, in which the clinical teachers reviewed the prerequisite knowledge and acquainted them with the expectations and details of roles and responsibilities for performing each task. Moreover, encounters with the tasks were systematically monitored to ensure correct performance of the tasks by the students.Combined ML–TBL method in the intervention group: The students of this group experienced the same teaching–learning process as the second control group. In addition, they received the five micro-videos through WhatsApp social media on the first day of the rotation. They were briefed to first refer to the videos to resolve their problems while performing the tasks, and then, if necessary, ask the clinical teachers or residents for further guidance.

### 2.5. Statistical Analysis

The data were analyzed using IBM SPSS Statistics for Windows, version 17 (IBM Corp., Armonk, New York, NY, USA). We used the Shapiro–Wilk test and shape of distribution for examining the normality of data distribution. In addition, we conducted a paired t-test, ANOVA and ANCOVA for analyzing parametric variables, and a Kruskal–Wallis test for non-parametric ones.

## 3. Results

All 59 students took the pre- and post-tests. There were no significant differences among the study groups with regard to the gender and age (Table 1).

Table 2 shows the comparison of the knowledge scores within and among three study groups. In all cases, the normal distribution of the samples was checked using the Shapiro–Wilk test, shape of samples distribution, QQ plot, kurtosis and skewness. The results showed that while there were no significant differences in knowledge pre-test scores among three groups (*p*-value = 0.628), significant increases were observed between the pre- and post-test scores in each group (*p*-values of 0.001, 0.024 and 0.001 for control 1, control 2 and intervention groups respectively). We used analysis of covariance to compare the post-test scores among three groups with elimination of the effect of the pre-test scores. To do so, we considered the study groups, post-test scores and pre-test scores as independent variable, dependent variable and covariate, respectively. The results showed a significant difference among the post-test scores (F = 3.423, *p*-value = 0.040).

We applied the Kruskal–Wallis test for comparing the mean DOPS scores among study groups due to non-normal distribution of scores in the second control group. As indicated in Table 3, the DOPS scores in the intervention group are higher than the control ones for all the tasks.

We conducted the Repeated Measurement ANOVA test to assess the differences of mean DOPS scores for each task in the study groups and observed no significant differences (Table 4 and Figure 1). This finding indicates that the task type had no effect on the results.

## 4. Discussion

In this study, we assessed the effect of the integrated ML–TBL strategy on the knowledge and performance of final-year medical students in their otolaryngology rotation. In order to be able to measure the effect of the integrated approach, the study was conducted in three groups, including two control (group 1: routine teaching and group 2: TBL methods) and one intervention (integrated ML–TBL method) group. The result revealed significant increases in participant knowledge and performance in the intervention group compared to the control ones.

ML has some advantages that we believe were influential in our study. The asynchronistic nature of ML provides the students with the possibility of controlling the place and time of learning. This characteristic, alongside with the short length and just-to-point content, make ML a suitable strategy for learning quickly within the minimum time span, i.e., just-in-time learning. It allows students to have access to the information in the moment that they need to learn with the help of technology [5] and improves their levels of cognition and skills [23,24]. These advantages can be useful in clinical education, where there is patient load [15] and students need to have access to relevant and concise information while managing patients. There are some studies that have assessed the effect of ML in medical education, though a few have assessed students’ clinical performance in a real work setting, and most of them have evaluated students’ reaction and knowledge acquisition [5]. ML alone is not recommended for teaching complex tasks in real work environments and it should be implemented in combination with other instructional strategies that are appropriate for learning in such settings [7].

In this study, we combined ML with TBL, because medical students have to learn tasks that are expected to be performed in their future real job. This requires adopting clinical teaching methods that are appropriate for workplace-based learning, among which TBL is a recommended approach. This strategy provides the chance of just-in-time training while performing a task. To implement this strategy, after selecting and designing the tasks, the support information for students’ learning should be identified and provided in a concise and just-to-point way [25]. We covered this step by providing students with micro videos. Our results are supported by Cheng et al.; they compared the use of a just-in-time micro-video with reading textbooks for performing the task of splinting technique and found that watching a three-minute video immediately before performing the technique resulted in shorter preparation time, higher performance assessment scores and a higher rate of successful splint application in comparison with reading medical textbooks [26]. We observed the same finding and significant increases in DOPS scores revealed better task performance in the intervention group who had access to micro-videos. We could not find any other study focusing on an integrated ML–TBL approach.

However, there are other studies that have separately examined the effect of learning through ML or TBL in medical education. Most of the studies have addressed ML effect on participants’ learning (the second level in Kirkpatrick model) and evaluated knowledge and not skill acquisition. Their results showed higher knowledge scores with the help of ML [5]. In one study, providing surgery clerkship students with a ML module resulted in higher knowledge scores compared to the control group [27]. These results support the present study findings on improvement of knowledge scores in the intervention group.

Meanwhile, Tian et al. [27] compared the effect of TBL and conventional lecture-based method with regard to the theoretical and practical scores of postgraduate medical students in a course on immunohistochemistry. They found significant differences in the mean score of the practical test, in contrast to the theoretical test scores which showed no significant differences. The authors indicated that lecture-based learning could transfer knowledge to the students systematically and comprehensively, although it was insufficient for practical problem solving while performing laboratory exercises. On the other hand, TBL is effective for problem solving while performing the tasks, because of greater student engagement compared to lecture sessions [27]. In our experience, we achieved the same results and mean knowledge score was higher in the first control group than the second one.

Finally, there is a need to decide on an appropriate microlearning method for different educational subjects [9]. In our experience, students’ DOPS scores increased for all the tasks regardless of their types which covered physical examination, patient management and clinical procedures. This finding, alongside with the study by Cheng et al. [26], indicates that micro-videos being accessible in clinical setting enhance learners’ performance in conducting a variety of medical tasks. We assume that ML is especially helpful in clinical disciplines like otolaryngology, where medical students are the first line encounters of the patients in emergency units and clinics and there is the challenge of time constraint. On the other hand, we suggest further research on usefulness of ML in clinical contexts with more flexibility, where students have more time for learning prerequisite knowledge and then performing clinical tasks.

We recognize some limitations for our study. It was only implemented in a small number of clerkship students on a two-week otolaryngology rotation in one hospital. Hence, whether our findings are transferable to other clinical education settings is an area for future research. We assessed the second level of the Kirkpatrick model including both knowledge acquisition and task performance. Moreover, among different ML modalities and techniques, we implemented micro-videos. We recommend future research in this field should address higher levels of learning outcomes from various ML modalities.

## 5. Conclusions

ML could have an important place in clinical education especially when integrated with other clinical education strategies like TBL. ML facilitates students’ learning through concise, short learning units which are integrated into daily routine and accessed on-demand while performing the tasks. Such combined strategies can enhance both knowledge and skill acquisition in medical students.

## Figures and Tables

**Figure 1 ijerph-20-04489-f001:**
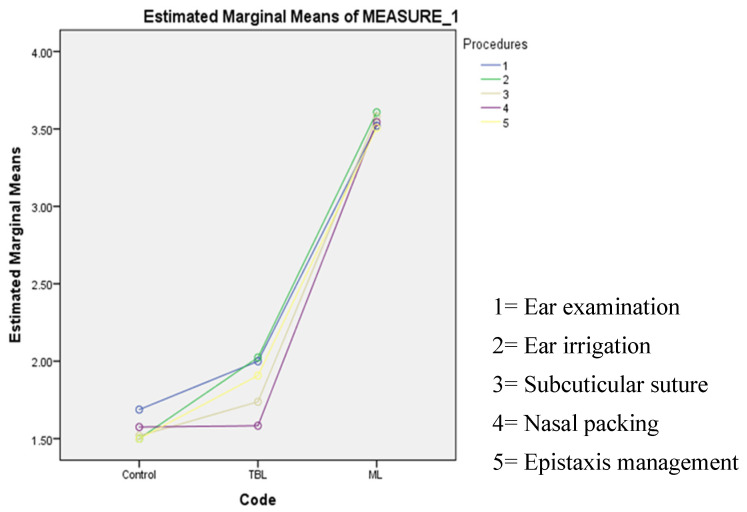
The result of Repeated Measurement ANOVA test for comparing the mean DOPS scores for each task in three study groups.

**Table 1 ijerph-20-04489-t001:** Comparison of participants’ gender and age among the study groups.

Group	N	Gender			Age		
		Female (%)	Male (%)	Sig *	Mean year (SD)	F	Sig **
Control 1	20	14 (70%)	6 (30%)	0.52	25.35 (0.565)	0.266	0.767
Control 2	20	13 (65%)	7 (35%)		25.15 (1.348)		
Intervention	19	12 (63.2%)	7 (36.8%)		25.47 (1.264)		

* Chi square, ** independent samples *t*-test.

**Table 2 ijerph-20-04489-t002:** Within and among group comparison of the knowledge pre- and post-test scores.

Group	Knowledge Scores Out of 15 Mean (SD)		Sig *	ANCOVA		
	Pre-Test	Post-Test		F	Sig	Observed Power
Control 1	8.55 (1.87)	10.45 (1.46)	0.001	3.423	0.040	0.619
Control 2	9.05 (1.98)	10.15 (1.38)	0.024			
Intervention	8.57 (1.57)	11.00 (1.73)	0.001			
Sig **	0.628	0.156				

* Paired *t*-test, ** analysis of variances.

**Table 3 ijerph-20-04489-t003:** Comparison of DOPS scores for each task among the study groups.

Task	Group	Mean Out of 5 (SD)	Sig *
Ear examination	Control 1	1.59 (0.33)	0.001
	Control 2	1.81 (0.41)	
	Intervention	3.43 (0.36)	
	Total	2.26 (0.90)	
Ear irrigation	Control 1	1.44 (0.16)	0.001
	Control 2	1.79 (0.31)	
	Intervention	3.54 (0.21)	
	Total	2.36 (0.96)	
Subcuticular suture	Control 1	1.44 (0.20)	0.001
	Control 2	1.66 (0.23)	
	Intervention	3.57 (0.13)	
	Total	2.23 (0.98)	
Nasal packing	Control 1	1.52 (0.25)	0.001
	Control 2	1.58 (0.19)	
	Intervention	3.54 (0.12)	
	Total	2.39 (1.02)	
Epistaxis management	Control 1	1.43 (0.25)	0.001
	Control 2	1.72 (0.33)	
	Intervention	3.44 (0.26)	
	Total	2.16 (0.92)	

* Kruskal–Wallis.

**Table 4 ijerph-20-04489-t004:** Comparison of the mean DOPS scores for each task in three study groups using Repeated Measurement ANOVA test.

	Source	Type III Sum of Squares	df	Mean Square	F	Sig.
Greenhouse–Geisser	Control 1	0.23	1.793	0.11	1.29	0.30
	Control 2	0.83	2.31	0.36	3.14	0.76
	Intervention	0.11	1.75	0.63	1.53	0.23

## Data Availability

Not available.

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
