# Peer review of "Enhancing Medical Students’ Knowledge and Performance in Otolaryngology Rotation through Combining Microlearning and Task-Based Learning Strategies"

_ijerph, 2023, doi:10.3390/ijerph20054489_

Round 1

Reviewer 1 Report

This is an interesting paper dealing with the combination of Microlearning and Task-Based Learning Strategies using a prospective trial. The paper reads very well, methodology and results are clearly presented and straigtforward. The topic is discussed with all relevant literature. 

Ther are only two issues that needs some attendtion:

1. Despite the low number of students in each of the strata the results are statistically sound. Nevertheless, the question remains if the combination of Microlearning and Task-Based Learning Strategies ist also relevant for other disciplines besides otolaryngoscopy, or in which disciplines it would make no sense. Such a kind of guideline would be of interest.

2. Is Microlearning the most important part of the combination of Microlearning and Task-Based Learning Strategies. It would have been a good idea to have had 4 strata, the used 3 ones with the addition of a strata only using Microlearning without Task-Based Learning Strategies.

Author Response

On behalf of the authors, I would like to appreciate the reviewer 1 precious comments that helped us to improve the manuscript. We worked on the comments that are all listed and answered in the following and revised the manuscript precisely according to them. The revisions are addressed with line numbers and marked up in the attached manuscript file using the track changes function.  

Comment 1.

“Despite the low number of students in each of the strata the results are statistically sound. Nevertheless, the question remains if the combination of Microlearning and Task-Based Learning Strategies is also relevant for other disciplines besides Otolaryngology, or in which disciplines it would make no sense. Such a kind of guideline would be of interest.”

Explanation: This point had been explained in the manuscript, though we further elaborated the idea in lines 309 to 314 of the revised manuscript.

Comment 2.

“Is Microlearning the most important part of the combination of Microlearning and Task-Based Learning Strategies. It would have been a good idea to have had 4 strata, the used 3 ones with the addition of a strata only using Microlearning without Task-Based Learning Strategies.”

Explanation: We could not assign a stratum for using only microlearning because we had no choice of omitting the routine teaching strategy and substituting it with microlearning alone.  Routine teaching method in the ward was learning by conducting the assigned tasks under the supervision of residents and clinical teachers. Although this strategy was not TBL, it was similar to it to some extent. So, if we had assigned a group for microlearning, the results would have been diluted when comparing “routine teaching - microlearning” and “microlearning- task based learning” groups. To address this issue, we explained the teaching method before conducting this study with more details in lines 94 to 99 of the revised manuscript.

Reviewer 2 Report

This study aims at assessing the effect of the integrated approach of microlearning with task-based learning on medical students’ knowledge and performance in Ear, Nose, and Throat clerkship rotation. This study is considered meaningful because it reveals useful learning strategies in medical education. The manuscript is well-written and organized. The discussion is also richly written based on the research results. However, there are a few things to fix.

Read the [Instruction for the author] section of this journal carefully. The abstract should be written without headings. Delete headings like 'Background, methods, results, conclusion'.

Ethically, students are a vulnerable group. Although approved by the IRB, it is necessary to write in detail about how to ethically protect the research participants. How did the researcher ensure students' fair participation in class? Did students participate in the study voluntarily? How were students rewarded for participating in the research?

In the Data collection section, you need to write the specific date the study was done.

The clinical teachers created a micro-video for each of the above-mentioned tasks. How can we ensure the validity and standard of the developed Micro-video?

Author Response

On behalf of the authors, I would like to appreciate the reviewer 2 precious comments that helped us to improve the manuscript. We worked on the comments that are all listed and answered in the following and revised the manuscript precisely according to them. The revisions are addressed with line numbers and marked up in the attached manuscript file using the track changes function.

Comment 1.

“Read the [Instruction for the author] section of this journal carefully. The abstract should be written without headings. Delete headings like 'Background, methods, results, conclusion'. “

Explanation: We edited the abstract and omitted the headings.

Comment 2.

“Ethically, students are a vulnerable group. Although approved by the IRB, it is necessary to write in detail about how to ethically protect the research participants. How did the researcher ensure students' fair participation in class? Did students participate in the study voluntarily? How were students rewarded for participating in the research?”

Explanation: We explained the ethical issue with more details in lines 102 to 105 of the track changed manuscript file.

Comment 3.

“In the Data collection section, you need to write the specific date the study was done.”

 Explanation: We specified the date in lines 87 to 88 of the track changed manuscript file..

Comment 4.

“The clinical teachers created a micro-video for each of the above-mentioned tasks. How can we ensure the validity and standard of the developed Micro-video?”

 Explanation: We explained this concern in lines 119 to 122 of the track changed manuscript file..
